



# Improved Bathymetry Estimates Beneath Amundsen Sea Ice Shelves using a Markov Chain Monte Carlo Gravity Inversion (GravMCMC, version 1)

Michael J. Field[1], Emma J. MacKie[1], Lijing Wang[2], Atsuhiro Muto[3], and Niya Shao[1]

[1]Department of Geological Sciences, University of Florida, 241 Williamson Hall, Gainesville, FL, 32611
[2]Department of Earth Sciences, University of Connecticut, Beach Hall Room 207, 354 Mansfield Road - Unit 1045, Storrs, CT 06269
[3]Department of Earth and Environmental Science, Temple University, 1901 N. 13th Street, Philadelphia, PA, 19122

**Correspondence:** Michael J. Field (mjfield2@outlook.com)

**Abstract.** Bathymetry beneath ice shelves in West Antarctica plays an important role in the delivery and circulation of warm waters to the ice-shelf bottom and grounding line. Large-scale bathymetric estimates can only be inferred through inversion of airborne gravity measurements. However, previous bathymetry inversions have not robustly quantified uncertainty in the bathymetry due to assumptions inherent in the inversion, and typically only produce a single model, making it difficult to prop-

agate uncertainty into ocean and ice-sheet models. Previous inversions have sometimes considered uncertainties in bathymetry models due to background densities but have not quantified the uncertainty due to the non-uniqueness inherent in gravity and geological variability below ice shelves. To address these issues, we develop a method to generate ensembles of bathymetry models beneath the Crosson, Dotson, and Thwaites Eastern ice shelves with independent realizations of background density and geological variability, represented through the Bouguer gravity disturbance. We sample the uncertainty in the unknown

geology below the ice shelves by interpolating the Bouguer disturbance using Sequential Gaussian Simulation. Each inversion is efficiently solved using a random walk Metropolis Markov Chain Monte Carlo approach which randomly updates blocks of bathymetry and accepts or rejects updates. Our ensembles of bathymetry models differ from previous estimates of bathymetry by hundreds of meters in some areas and show that the uncertainty in the Bouguer disturbance is the largest source of uncertainty. The different bathymetry models in the ensembles can be used in oceanographic models to place better bounds on

sub-ice-shelf melting and future grounding line retreat.

## 1 Introduction

Introduction and Background West Antarctica is experiencing rapid ice-shelf thinning and increases in ice velocity, driven largely by the delivery of warm modified Circumpolar Deep Water (mCDW) to the ice-shelf bottom (Mouginot et al., 2014; Steig et al., 2012; Rignot et al., 2014; Milillo et al., 2019; Rignot et al., 2024). The amount of melting at the bottom of the ice

shelves is controlled by the circulation of water, which is in turn controlled by the shape of the sub-ice-shelf cavities (Seroussi et al., 2017; Steig et al., 2012). For these reasons, knowledge about the geometry of the water column below the ice shelves, or the bathymetry, is critical to better projecting the future evolution of the ice shelves and the glaciers they buttress. Additionally,





uncovering sub-ice-shelf bathymetry can tell us about previous pinning points and patterns of destabilization (Tinto and Bell, 2011; Favier et al., 2012; Docquier et al., 2014). The critical role that sub-ice-shelf bathymetry and subglacial topography

play in grounding-line retreat in West Antarctica means that it is essential to produce accurate estimates of bathymetry and to quantify the uncertainty of these estimates in a way that their influence on ice-sheet models can be better understood (MacKie et al., 2021).

Although the thickness of the ice shelves can easily be measured using Radio Echo Sounding (RES) (e.g., Schroeder, 2023), measuring the depth of the seafloor beneath the ice shelves remains difficult because radio waves cannot penetrate the

seawater column. Other measurement methods, such as seismic sounding and underwater autonomous or remotely operated vehicles, remain costly and sparse in practice (e.g., Muto et al., 2016; Schmidt et al., 2023). For these reasons, the primary instruments used to infer sub-ice-shelf bathymetry are airborne gravity systems, which allow for rapid data acquisition over large areas. Airborne gravimeters measure the acceleration due to gravity and are processed into gravity anomalies that reflect density anomalies in the subsurface (Blakely, 1995). Airborne gravity surveys can cover hundreds of square kilometers with

flight line spacing typically greater than 5 km (e.g., Tinto et al., 2010; Jordan et al., 2020b). Gravity inversions seek to infer the distributions of densities which give rise to the observed gravity anomalies (Blakely, 1995). The large density contrast between rock and water means that large-scale variations in seafloor topography produce a significant signal. However, gravity inversions are fundamentally non-unique, meaning that there are many different density contrasts and bed geometries which could reproduce the observed gravity anomalies (Blakely, 1995). Because of this, it has been typical to fix the density of rock

volumes in the model and invert only for the bathymetric surface (e.g., Tinto and Bell, 2011; Jordan et al., 2020b). Even with fixed densities, the non-uniqueness of the problem is such that multiple bathymetry models can reproduce the gravity data within the measurement uncertainty. For example, a sub-ice-shelf cavity could be deeper with a greater bedrock density, or the cavity could be shallower with a lower bedrock density (Brisbourne et al., 2014). Additionally, one small area could be shallower, and an adjacent area could be deeper and produce the same gravity anomaly.

A commonality among gravity inversions for bathymetry in Antarctica and Greenland is the correction of the difference between the observed free-air anomaly and forward modeled gravity to pin the inversion to known bed elevations (Jordan et al., 2020b; Hodgson et al., 2019; Tinto et al., 2015; Boghosian et al., 2015; Charrassin et al., 2025). This usually entails forward modeling the gravity due to ice, water, and rock (the terrain effect) using an initial model, and interpolating the difference between the free-air anomaly and terrain effect conditioned on where the bedrock topography is well constrained. This interpo-

lation is done with a spline or similar deterministic interpolation (Jordan et al., 2020b; An et al., 2019), or sometimes a constant offset (Millan et al., 2017; Tinto and Bell, 2011). The interpolated residual field is then subtracted from the free-air anomaly and this terrain effect is used as the target of the inversion. Over areas of known elevation, the residual gravity reflects deviations from the forward modeled density and crustal thickness, making it a complete Bouguer disturbance (see Section 2 for a note on our use of the term disturbance as opposed to anomaly). The inversion is then essentially finding the terrain effect which,

when added to the complete Bouguer disturbance, equals the observed gravity disturbance. In this paper, we will refer to this residual gravity interpolation as the Bouguer disturbance to highlight the fact that this field represents crustal density anomalies and crustal thickness. The fundamental issue facing sub-ice-shelf bathymetry inversions is that neither the terrain effect nor the



Bouguer disturbance are known at the ice shelves, and the non-uniqueness between the two is such that the interpolation of the Bouguer disturbance can result in large errors (Brisbourne et al., 2014). As such, the deterministic interpolations previously employed place strong assumptions on the sub-ice-shelf geology and result in an inadequate representation of uncertainty. In this paper, we will discuss how geostatistical interpolations of the Bouguer disturbance over areas of unknown bed elevation can be used to incorporate the uncertainty in sub-ice-shelf geology into the bathymetry uncertainty.

Previous studies have estimated the sub-ice-shelf bathymetry of ice shelves in the Amundsen Sea Embayment (ASE). Tinto and Bell (2011) provided the first gravity-derived estimates of sub-ice-shelf bathymetry at Thwaites Glacier using Operation Ice Bridge (OIB) data (Tinto et al., 2010). This study modeled multiple gravity lines in 2D using the forward gravity method of Talwani et al. (1959), which computes the gravity due to an arbitrary polygon on a 2D surface. Where a poor fit was achieved, a higher density rock was used to improve the fit. After the bathymetry was estimated along each flight profile, these bathymetry profiles were interpolated to produce a continuous bathymetric surface. This study resulted in the identification of a prominent ridge offshore of Thwaites which could have acted as a previous pinning point and could play an important role in water circulation to the grounding line. This study was an important first step for modeling sub-ice-shelf bathymetry in the ASE, but interpolating independent 2D inversions neglects the effects of off-track geometry and likely introduces error, which cannot be captured in uncertainty estimates. Jordan et al. (2020b) provided updated bathymetric estimates for the area around Thwaites, Crosson, and Dotson ice shelves in the ASE using OIB and International Thwaites Glacier Consortium (ITGC) airborne gravity data. The authors used a 3D modeling approach with a deterministic inversion to produce an initial bathymetric estimate. They then refined their model using a "topographic shift" method, wherein error between the initial estimate and known bed elevations derived from ice penetrating radar or Multi-Beam Echo Sounding (MBES) was interpolated and subtracted from the initial estimate. Uncertainty was estimated by re-running the inversion while excluding shipborne MBES data in a small portion of the model. The standard deviation of the resulting errors was 100 m. This analysis provides valuable information about the benefits of incorporating the additional dataset, but does not provide information about how uncertainty in the bathymetry varies across space. Millan et al. (2017) estimated the sub-ice-shelf bathymetry beneath ASE ice shelves using an iterative inversion method but lacked the ITGC data and robust spatial uncertainty quantification. Additionally, new MBES data acquired in the Amundsen Sea showed that Millan et al. (2017) produced errors up to 150 m due to small scale geomorphological features (Hogan et al., 2020). Given the complexity of the seafloor in the Amundsen Sea embayment shown by Hogan et al. (2020), errors on the order of hundreds of meters for a single model are expected and further motivates robust uncertainty quantification. Recently, Charrassin et al. (2025) produced new bathymetry maps for ice shelves around Antarctica using the topographic shift method of An et al. (2019). Their work produced a spatially variable uncertainty map based on the misfit between the forward modeled and observed gravity using a conversion of 5.8 mGal per hundred meters of bathymetry. The authors also quantified uncertainty by holding out sonar and seismic data. However, this recent bathymetry compilation produced only a single estimate of bathymetry for each domain and did not quantify uncertainty due to sub-ice-shelf geologic variability (Charrassin et al., 2025). Finally, many of the previous studies use proprietary software, which limits examination and reproduction of the work (e.g., Jordan et al., 2020b).





Other studies have used more robust statistical optimization techniques to provide estimates of subglacial features along with uncertainty estimates using airborne gravity data. Muto et al. (2016) used Very Fast Simulated Annealing (VFSA), a variant of simulated annealing, to invert for on-shore and off-shore bathymetry at Pine Island Glacier (PIG) in the ASE and Roy et al. (2005) used VFSA to invert for the subglacial-lake geometry of Lake Vostok. VFSA is a global optimization method similar to Markov Chain Monte Carlo (MCMC), in which proposed updates to model parameters are accepted or rejected using a criterion based on how well the forward-modeled data fits the observed data (Ingber, 1989; Sen and Stoffa, 2013). VFSA converges to a solution by proposing successively smaller updates to model parameters while calibrating the search size independently for each model parameter. Both studies inverted for basement and sediment thickness geometry simultaneously and provided uncertainty bounds for both. However, these uncertainty bounds are not directly useful because the bathymetry and sediment thickness are non-unique. Without realizations of the bathymetry and sediment thickness, the uncertainty cannot be propagated into ice-sheet and ocean models. Additionally, these studies did not probe other assumptions inherent in the gravity inversion, such as the assumption of background density and interpolation of the Bouguer disturbance. The interpolation of the Bouguer disturbance has previously not been considered as a contributor to uncertainty, and thus the effects of the uncertain interpolation on the resulting bathymetry and ice-sheet and ocean models that use the bathymetry have not been explored. As such, a new probabilistic inversion approach that can produce different realizations of bathymetry such that the uncertainty can be propagated to ice-sheet and ocean models is required.

Previous sub-ice-shelf bathymetry inversions did not produce multiple realizations. Tinto and Bell (2011), Jordan et al. (2020b), and Millan et al. (2017) each provided a single model with uncertainty estimates of 70 m, 100 m, and 50 to 65 m, respectively. Tinto and Bell (2011) estimated uncertainty based on uncertainty in the gravity data and by varying the background density. Jordan et al. (2020b) estimated uncertainty by holding out conditioning data, and Millan et al. (2017) estimated uncertainty using the misfit of between the modeled and observed gravity data. These analyses are beneficial for giving an impression of how accurate the results might be, but with only a single bathymetry model from each inversion, there is no way to visualize different possible bathymetry models and apply them to ocean and ice-sheet models. Additionally, it is unclear if sensitivity analyses of the background density have carried the assumption through the whole inversion, including the initial forward modeling used to interpolate the Bouguer disturbance. Thus, with unclear treatment of the assumptions throughout the whole inversion and with only single-value estimates of uncertainty, the utility of such uncertainty estimates is unclear beyond providing a general sense of the uncertainty of the model.

The uncertainty of bathymetry across space can more easily be captured through an ensemble of model realizations from a stochastic process. The importance of ensembles of stochastically simulated subglacial topography for uncertainty quantification has already been demonstrated. For example, MacKie et al. (2020) showed that different realizations of geostatistically simulated bed topography can produce different spatial distributions of subglacial lakes. Similarly, MacKie et al. (2021) demonstrated that subglacial water routing pathways are sensitive to slight topography variations that result from geologically realistic simulation of bed topography. Finally, Wernecke et al. (2022) showed how an ensemble of geostatistically simulated bed topographies could be used to propagate topographic uncertainty to uncertainty in future sea-level contributions from Pine Island Glacier. Each of these examples demonstrate how realizations of a stochastic processes which honor the spatial statistics of the



underlying conditioning data enable more robust uncertainty quantification of ice-sheet processes. Previous bathymetry inversions have lacked a stochastic inversion process which would enable such generation of bathymetry realizations and rigorous uncertainty quantification.

One approach to generating multiple solutions to an inverse problem is MCMC. The general strategy of MCMC is to move throughout a parameter space by proposing moves to new model parameters and accepting these moves with a probability based on the fit between the observed and forward-modeled data. For example, Fu and Gómez-Hernández (2009) used MCMC to sample a hydraulic-conductivity field inversely conditioned on hydraulic-state data. MCMC has been used extensively in seismic inversions of reservoir properties. See Grana et al. (2022) for an extensive review. Wang et al. (2023) used MCMC

and explicit and implicit surface perturbations to invert for a variety of geological interfaces. The benefits of these sampling approaches are that only a forward model and some mechanism to propose new models are required. These approaches are becoming more popular because of their ability to solve inverse problems by generating samples from a posterior distribution such that uncertainty can be quantified with an ensemble of model realizations while honoring conditioning data. As such, an MCMC approach could be used to sample solutions to the gravity inverse problem while honoring gravity constraints.

Up to this point, previous gravity inversions have not provided robust uncertainty quantification, making it difficult to propagate the uncertainty in bathymetry to ice-shelf-melt processes (Jordan et al., 2020b; Tinto and Bell, 2011; Millan et al., 2017). In order to fill this need, we developed a stochastic inversion approach which leverages geostatistical simulation to interrogate the assumptions inherent in the inversion while providing models ready for input into ocean and ice-sheet models such that the uncertainty can be propagated to ice sheet and ocean processes. We generate three different ensembles of bathymetry mod-

els that sample the uncertainty in the inversion in different ways, allowing for comparison of the sources of uncertainty. The first ensemble includes sampling of the background density and geostatistical interpolation of the Bouguer disturbance. The second ensemble uses a constant background density and geostatistical interpolation of the Bouguer disturbance, and the final ensemble uses a constant background density and a deterministic interpolation of the Bouguer disturbance. The geostatistical interpolation uses Sequential Gaussian Simulation (SGS) to generate realizations of the Bouguer disturbance, representing dif-

ferent versions of sub-ice-shelf geological variability. We perform each inversion using a modified Random Walk Metropolis MCMC algorithm that converges to a bathymetry model by randomly updating blocks of the bathymetry. These ensembles of bathymetry models represent the most robust quantification of uncertainty in sub-ice-shelf bathymetry inversions to date and are ready to be used in ocean and ice-sheet models. We also discuss the differences between our ensembles and previous estimates, and the possibilities that our ensembles present for better understanding basal melt processes, future ice sheet melting,

and ice-sheet history.

## 2  Data

The gravity data used in this study consists of Operation Ice Bridge (OIB) and International Thwaites Glacier Collaboration (ITGC) airborne gravity data. The OIB data were collected on a multi-instrument mission between 2009 and 2011. The gravimeter was a Sander AIRGrav inertial platform system flown on a DC-8 aircraft typically traveling at 120 m/s at 450 m



**Figure 1.** (a) Study area in the Amundsen Sea Embayment. Ice velocity in log scale (Mouginot et al., 2019a) overlain on the 2014 Mosaic of Antarctica satellite image (Scambos et al., 2007; Haran et al., 2017). The extent of the grounding zone is shown with pink lines (Rignot et al., 2023).(b) Coverage of gravity data in the study region. The orange shows ITGC flightlines and the purple shows OIB flightlines (Tinto et al., 2010; Jordan et al., 2020b). (c) Coverage of sonar and radar data in the study region from BedMachine v3 (Morlighem et al., 2020).

altitude above the ice surface with a typical line spacing of about 10 km (Tinto et al., 2010; Tinto and Bell, 2011; Sander et al., 2004). We used the free-air anomaly data filtered with a 70-second filter representing a half-wavelength resolution of 5.2 km. The resulting free-air anomaly has crossover errors of about 1.67 mGal (Tinto et al., 2010). The ITGC data were collected between 2018 and 2019 using a DEO iMAR RQH 4001 Inertial Navigation System strapdown system flown on a Twin Otter aircraft traveling at  60 m/s with a typical altitude of 340 m above the ice surface and a typical line spacing of 7.5 km (Jordan



et al., 2020b). We used the free-air anomaly data filtered to 5 km resolution. The resulting data has a crossover error of 1.56
mGal.

In both gravity datasets, the gravity anomaly is reported as a free-air gravity anomaly, but we refer to them as gravity distur-
bance given that the free-air correction is referenced to the ellipsoid, which aligns with the notion of gravity disturbance used in
geodesy (see e.g., Hackney and Featherstone, 2003). Throughout this paper, we will refer to the terrain gravity disturbance and

the complete Bouguer gravity disturbance. The terrain effect refers to all anomalous mass relative to the ellipsoid, including
ice, water, and rock above the ellipsoid, and air below the ellipsoid. The terrain effect is also analogous to a complete Bouguer
correction which, when subtracted from the observed gravity disturbance, produces the complete Bouguer disturbance. The
Bouguer disturbance constitutes a large proportion of the observed gravity disturbance arising from anomalous bodies within
the crust and variations in the crustal thickness.

We used the BedMachine v3 continent-wide gridded product (Morlighem et al., 2020) for the bed elevation outside of the
inversion domain, the ice surface, and ice thickness. The BedMachine bed outside of the inversion domain is assumed to be
accurate because of the high density of RES flight lines on grounded ice, and Multibeam Echo Sounding (MBES) offshore
(Hogan et al., 2020) (see Fig. 1c). We regridded the BedMachine products to 2-km resolution using a block median reduction
from the open source Verde Python package (Uieda, 2018). In the case of discrete data such as the BedMachine mask, we took

the nearest value to the gridded data point. The regridded BedMachine bed, ice thickness and source as shown in Fig. 2. We
then removed the gravity data collected above 1200 m flight elevation and upward continued the data to 1200 m flight elevation
onto the same 2-km BedMachine grid using gradient boosted equivalent sources included in the open source Python gravity
modeling package Harmonica (Soler and Uieda, 2021; Fatiando A Terra Project et al., 2023). We chose to upward continue the
gravity data onto a grid because the gravity data are oversampled along flightlines (see Fig. 1b), and so that the geostatistical

simulation could be performed (see Section 3.2). For the gradient boosted equivalent sources we placed point sources below the
original gravity measurement locations and selected the damping and depth-to-source hyperparameters using cross validation,
wherein all combinations of dampings of 0.1, 1, 100 and depths of 1 km, 2 km, and 10 km were tested on a portion of the data
held out. The best performing parameter combination was depth of 3000 m and damping of 1. These parameters were then
used to fit the synthetic sources to the original gravity data and forward modeled the gravity disturbance onto the 2-km grid at

an elevation of 1200 m. Finally, we masked out grid cells in the new gravity grid which were more than 4 km from the original
gravity locations because of the decaying amplitude from the equivalent sources. The final preprocessed gravity disturbance is
shown in Fig. 3a.

Between December of 2019 and January of 2020, 27 seismic soundings were conducted in the Bear Island strait and 37
soundings on Thwaites Eastern Ice Shelf by the Thwaites Amundsen Regional Survey and Network (TARSAN) team as part of

the ITGC (Rocarro, 2020; Muto et al., 2024). These soundings were collected alongside seismic refraction surveys in order to
constrain the ice and firn velocities to ensure accurate inversion of water-column thickness. These seismic soundings provide
valuable constraints on the sub-ice-shelf bathymetry inversion and thus were incorporated into the conditioning data. The
seismic bed elevations were interpolated using cubic spline interpolation masked to within 2 km of the seismic soundings. We





determined the best damping parameter of the spline interpolator to be $10^{-4}$ using cross validation. The BedMachine bed was
then replaced with the masked interpolation of the seismic soundings.

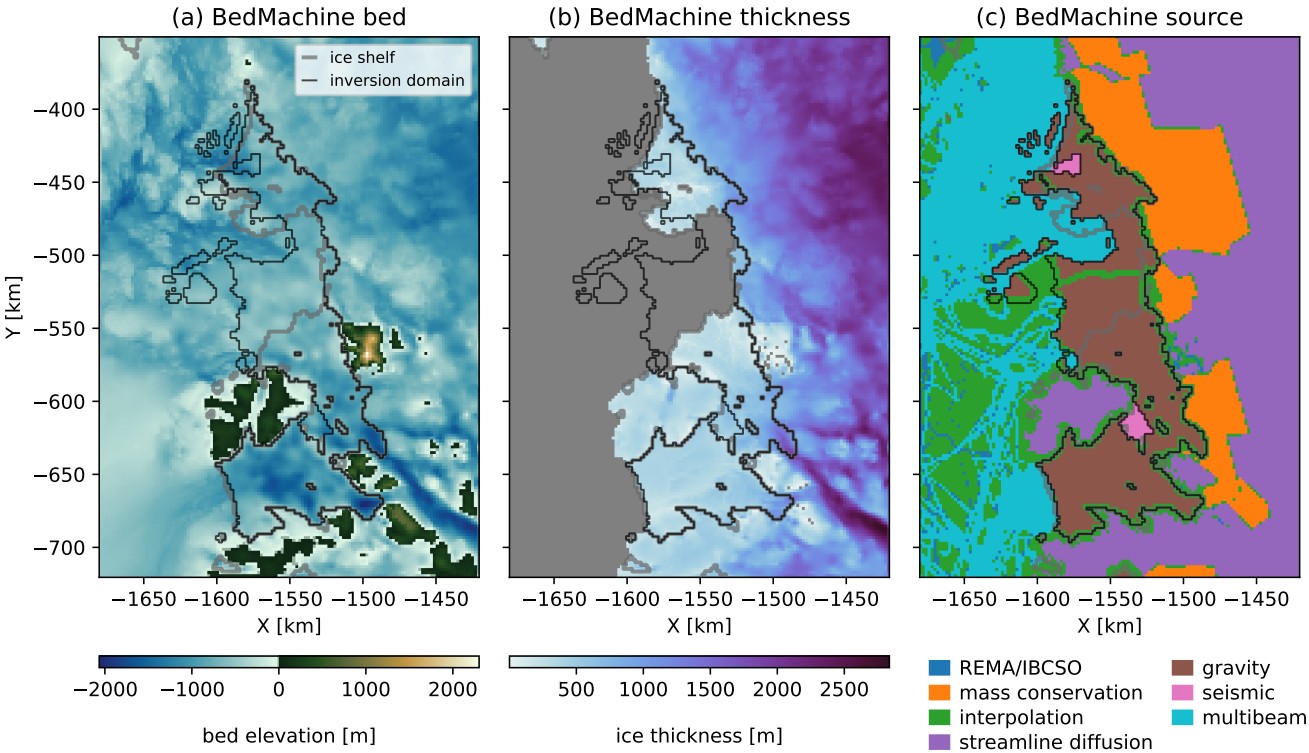

**Figure 2.** (a) BedMachine v3 bed elevation with respect to sea level. (b) BedMachine ice thickness (c). BedMachine source showing the
gravity sourced bathymetry in brown and bathymetry from seismic soundings in pink. Each panel has the inversion domain outline in black
and ice shelves outlined in gray.

# 3 Methods

## 3.1 Overview

To test the relative influences of the background rock density and Bouguer disturbance uncertainties on bathymetry estimates,
we ran three ensembles: one with variable background densities sampled from a Gaussian prior distribution centered at 2700
$\mathrm{kg/m^3}$ with a standard deviation of $80\ \mathrm{kg/m^3}$ and SGS interpolation of the Bouguer anomaly; one with constant background
density and SGS interpolation of the Bouguer anomaly, and finally; one with constant background density and a constant
Bouguer anomaly which is kriging interpolation of the Bouguer disturbance (Table 1). Each ensemble is composed of 100
inversions run in parallel on a 2021 Mac Studio M1 computer, each taking about 5 minutes to complete. The inversions are




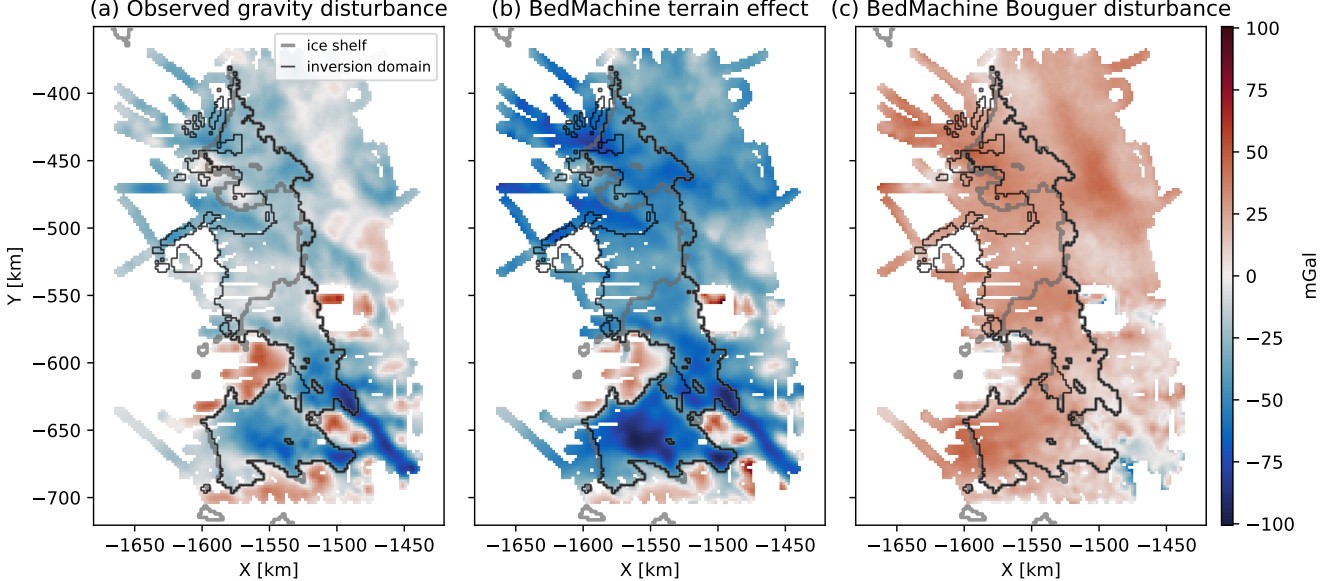

**Figure 3.** (a) Observed gravity disturbance. (b) Forward modeled gravity from anomalous ice, water, and rock relative to the ellipsoid using BedMachine. (c) The complete Bouguer disturbance from BedMachine, calculated as the difference between (a) and (b).

**Table 1.** Treatment of Bouguer disturbance and background density in different ensembles

| Ensemble number | Bouguer disturbance | Background density |
|---|---|---|
| 1 | Variable - SGS | Variable - $N(\mu = 2700, \sigma = 80)$ |
| 2 | Variable - SGS | Constant - 2670 $kg/m^3$ |
| 3 | Constant - Kriging | Constant - 2670 $kg/m^3$ |

completed using a random walk Metropolis MCMC approach that iteratively updates the bathymetry with block updates until

the root mean squared error (RMSE) between the forward modeled gravity and the target terrain effect is less than 1.2 mGal. This stopping criterion was chosen because the inversions were able to reliably achieve this RMSE. The final iteration of each inversion, the misfit throughout optimization, and the accepted iterations are saved at the end of each inversion. Fig. 4 describes the MCMC approach used for the inversions. The following sections describe how the Bouguer disturbance is stochastically interpolated with SGS, derive the random walk MCMC approach for this problem, explain the choice of block updates, describe

the forward modeling of the gravity data, and finally describe the initialization of the bathymetry models.

## 3.2 Geostatistical simulation of Bouguer disturbance

SGS is a geostatistical interpolation technique that preserves the spatial statistics of conditioning data while sampling the uncertainty of the interpolation (Deutsch and Journel, 1992). In contrast to SGS, traditional interpolation methods such as





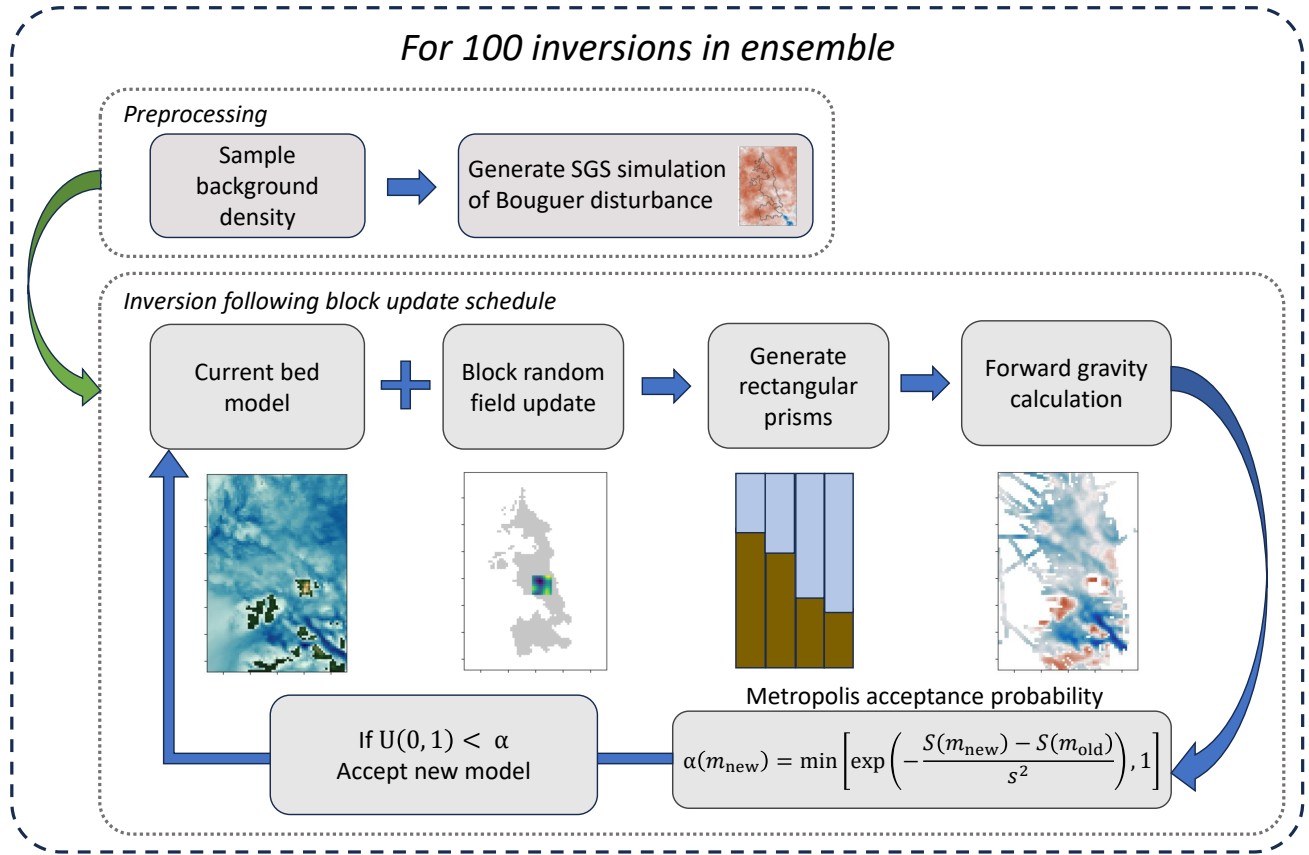

**Figure 4.** Flow chart showing the block update MCMC method for a single inversion.

kriging, splines, and Radial Basis Functions provide overly smooth interpolations that do not retain the spatial statistics of

the conditioning data. SGS works by moving through a random sequential path in the grid. At each grid cell, if the Bouguer

disturbance is unknown, a value is sampled from a Gaussian distribution defined by the kriging mean and variance and entered

into the grid (Deutsch and Journel, 1992). In this way, as the interpolation moves through the grid there are more and more

values from which to compute the kriging mean and variance, which is conditioned by a specified number of neighbors away

from the grid cell. The kriging equations incorporate the geostatistics of the conditioning data through the (semi)variogram,

which is a measure of the pairwise dissimilarity between data points with respect to the distance, or lag, between the pair of

data points. We generate the experimental variogram and fit Gaussian, exponential, spherical and Matérn covariance models to

the variogram using Scikit GStat (Mälicke, 2022). The Matérn covariance model fit the experimental variogram the best and

thus was used for the interpolation. The experimental variogram and fit are shown in Fig. 5b. We use the Ordinary Kriging

SGS interpolation function from the open-source Python package GStatSim (MacKie et al., 2023) where the Ordinary Kriging

SGS methods are robust to trends in the Bouguer disturbance. We generate 100 realizations of the SGS interpolation for use



in an ensemble that quantifies the uncertainty in the bathymetry due to the interpolation of the Bouguer disturbance. Finally, we interpolated the Bouguer disturbance deterministically with Ordinary Kriging using the same Matérn variogram (Fig. 5e) to create an ensemble without the uncertainty due to the interpolation.



**Figure 5.** (a) High confidence Bouguer disturbance data (see Fig. 1c for data coverage). (b) Semivariogram of Bouguer disturbance conditioning data. (c) SGS realization of Bouguer disturbance interpolation. (d) Same as (c). (d) The mean of 100 SGS interpolations. (e) The standard deviation of 100 SGS interpolations.





### 3.3 Annealed random walk Metropolis solution to inverse problem

We solve the gravity inverse problem using an MCMC approach where random blocks of the bed topography are iteratively perturbed until convergence is achieved. The approach works by iteratively proposing new bathymetric surfaces, forward modeling the gravity due to the new model, and accepting or rejecting the new model. After many iterations, the MCMC algorithm is able to converge to a bathymetry model that approximately matches the forward modeled gravity and the target terrain effect. An outline of the inversion process is shown in Fig. 4.

MCMC is a popular approach for generating samples from a posterior distribution of model parameters, often by proposing new models and accepting or rejecting moves to new models (Geyer, 2011). Typically, MCMC is viewed in a Bayesian context in which prior beliefs about the model parameters are transformed into the posterior distribution by a likelihood function which calculates the probability of a dataset given the model parameters (Tierney, 1994). However, our MCMC approach is best viewed as a stochastic optimization approach to generate a sample from the posterior within a larger Monte Carlo framework.

This is because the Bouguer disturbance and background density are sampled before the MCMC optimization and no prior probability density function is assigned to the bathymetry. This means that, ignoring the gravity constraints, each proposed bathymetry model is treated as equally probable in the MCMC. The effect of this is that each bathymetry inversion is driven solely by the likelihood function, or the fit to the gravity data, while the ensemble as a whole samples the prior distributions of the Bouguer disturbance and background density. Previous work has enforced geostatistical priors over model parameters

in MCMC in the form of SGS (e.g., Hansen et al., 2012; Mariethoz et al., 2010), however, we do not specify a prior for the sake of simplicity and to fit the gravity data as closely as possible. We use a random walk Metropolis (RWM) approach, a simplification of the popular Metropolis-Hastings MCMC algorithm, in which new model proposals are the previous model with the addition of a zero-mean multivariate normal perturbation (Hastings, 1970). In our case, these updates are 2D Gaussian random fields (see Section 3.4).

We derive the RWM gravity inverse problem following the notation of Mosegaard and Tarantola (1995). Let $m$ represent a vector of seafloor depth values. We aim to sample a posterior distribution, $\sigma(\mathrm{m})$, which, from Bayes' rule, has a solution of the form

$$\sigma(\mathrm{m}) = k\rho(\mathrm{m})L(\mathrm{m}) \tag{1}$$

where $L(\mathrm{m})$ is likelihood function evaluated on the bathymetry values, $k$ is a scaling factor, and $\rho(\mathrm{m})$ is a prior distribution

evaluated over the bathymetry values. As mentioned before, the bathymetry models are treated as equally probably and thus the prior can be ignored. We additionally define the forward model as

$$d = g(\mathrm{m}) \tag{2}$$

where $d$ is the gravity data and $g(\mathrm{m})$ is the forward gravity function evaluated on the bathymetry model. We assume independent and identically distributed Gaussian uncertainties $s$, making the likelihood function:





$$L(\mathrm{m}) = \exp\left(-\frac{S(\mathrm{m})}{s^2}\right) \tag{3}$$

where

$$S(\mathrm{m}) = \frac{1}{2}\sum_{i=1}^{N}\left(g^i(\mathrm{m}) - d_{\mathrm{obs}}^i\right)^2 \tag{4}$$

is the sum of squared residuals divided by two and $s$ is the uncertainty of the gravity data derived from crossover errors. The Metropolis-Hastings acceptance probability, $\alpha(\mathrm{m_{new}})$, is defined as

$$\alpha(\mathrm{m_{new}}) = \min\left[\frac{\sigma(\mathrm{m_{new}})q(\mathrm{m_{new}}, \mathrm{m_{old}})}{\sigma(\mathrm{m_{old}})q(\mathrm{m_{old}}, \mathrm{m_{new}})}, 1\right] \tag{5}$$

Where $q(\mathrm{m_{new}}, \mathrm{m_{old}})$ is the probability of proposing the new model given the old model. Given a symmetric proposal generation mechanism, as is the case with a zero-mean multivariate update, the probability of proposing the forward and reverse jumps are the same and the acceptance probability simplifies to

$$\alpha(\mathrm{m_{new}}) = \min\left[\frac{\sigma(\mathrm{m_{new}})}{\sigma(\mathrm{m_{old}})}, 1\right] \tag{6}$$

Since we cannot directly evaluate the target posterior distribution on the models, we can return to Bayes' rule and evaluate the unnormalized likelihoods of each model. The acceptance probability can then be put in terms of the misfit function as

$$\alpha(\mathrm{m_{new}}) = \min\left[\exp\left(-\frac{S(\mathrm{m_{new}}) - S(\mathrm{m_{old}})}{s^2}\right), 1\right] \tag{7}$$

This expression implies that if the new model improves the least squares misfit, the model is accepted and if it does not, there is still a non-zero probability that it is accepted. This simple approach is advantageous because we require no evaluation

of gradients and only require the forward gravity model and a multivariate normal perturbation method.

### 3.4 Block random field updates

The large number of bathymetry values to be modeled poses computational challenges in achieving convergence to the desired posterior distribution. In geostatistical inverse problems, using whole domain updates often results in very low acceptance rates and slow refinement of the model because one area may improve while another area worsens the fit with the data (Fu and

Gómez-Hernández, 2008). Because of this, it is advantageous to limit the updates to smaller portions of the model where the bathymetry is expected to be correlated in space (Fu and Gómez-Hernández, 2008; Reuschen et al., 2021).





**Table 2.** Block update parameter schedule

| Block width (pixels) | Range (pixels) | Amplitude (m) | Iterations |
|---|---|---|---|
| 21 | 10 | 60 | 1000 |
| 15 | 8 | 40 | 1000 |
| 9 | 6 | 40 | 5000 |
| 5 | 5 | 40 | 5000 |

To address this optimization challenge, we used block updates to perturb the bathymetry models. For each MCMC iteration, a random location within the inversion domain is selected as the center of the block. The block perturbation is a Gaussian random field confined to the block which is added to the previous bed.

Our ensemble approach relies on taking a sample from the ends of many MCMC chains, motivating a RWM approach which efficiently converges to producing samples within the distribution. In order to have the RWM converge as efficiently as possible we follow an annealing schedule which specifies the block size, the range and amplitude of the Gaussian random field, and number of iterations that the block update parameters are used for. The block update schedule used for the inversion is shown in Table 2. As the inversion converges to a bathymetry model, smaller block sizes with shorter ranges and lower amplitudes become more efficient. We tested different annealing schedules and qualitatively compared their acceptance rates and efficiency in reducing the gravity data misfit. We targeted acceptance rates between 0.2 and 0.5, which is an acceptable range for most MCMC applications (Rosenthal, 2011). The annealing schedule shown in Table 2 was chosen because it kept acceptance rates within the desired range and the reduction in RMSE per iteration high as seen in Fig. 6.

The block update method can introduce edge artifacts into the bathymetry. Many of these edges are smoothed out by the continued updating of the model, but some edges remain visible in the final iteration. To eliminate these features, the final iteration is low-pass filtered with a Gaussian filter with a cutoff wavelength of 5 km.

### 3.5 Gravity forward modeling

The gravity response, $d$, is forward modeled $g(m)$ using the gridded BedMachine v3 ice surface and ice thickness to generate rectangular prisms of constant density. We use an ice density of 917 $\mathrm{kg/m^3}$ and sea water density of 1027 $\mathrm{kg/m^3}$. We use the open-source Python package, Harmonica (Fatiando A Terra Project et al., 2023), to forward model the vertical gravitational acceleration of the prisms at the gridded gravity coordinates using the equations from Nagy et al. (2000). We then take the residual between the forward modeled data and the target terrain effect, calculate the Metropolis acceptance criterion (equation 6), and accept or reject the block update. The prism gravity method is advantageous because of its simplicity, but it can be computationally challenging because the gravity of each prism must be added together to compute the gravity at a single observation location. In order to speed up the gravity calculation, we leveraged the additive nature of the prism gravity method and limited the calculation to the updated block. For each gravity observation point, the gravity due to the new and previous prisms of the block are computed and the new gravity is calculated by subtracting the previous gravity due to the block prisms



and adding the gravity due to the new block prisms to the total gravity. This approach eliminates redundant gravity calculations and greatly speeds up the forward model calculation.

## 3.6 Initialization of the models

It is typical in MCMC to initialize chains with different initial conditions to avoid getting stuck in local modes (Geyer, 2011). Each inversion is initialized with the BedMachine bed with the addition of a Gaussian random field conditioned on the margins of the inversion domain so that each MCMC chain has a different initial condition. The primary and secondary range, or correlation distance, of the random field were both randomly sampled from uniform distributions with a maximum of 30 and 50 km to add anisotropy to the random fields. The random field is additionally scaled by a random value between 1 and 300 m and finally is conditioned on the margins of the inversion domain using Radial Basis Functions. These Gaussian random fields are generated using the open-source Python package GSTools (Müller et al., 2022).

## 4 Results

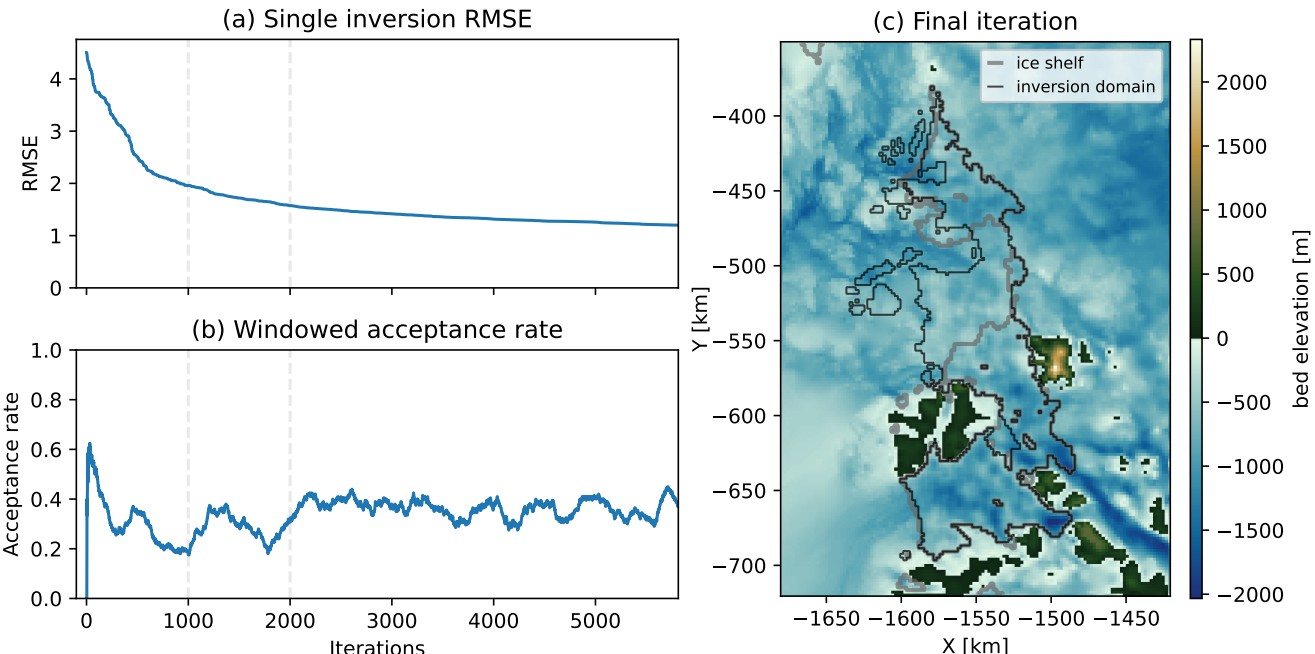

**Figure 6.** (a) RMSE between target terrain effect and forward modeled gravity. Vertical gray dashed bars indicate change in block update. (b) Running acceptance rate with 500 iteration window. (c) Bed elevation from final iteration of chain.

The RMSE, running acceptance rate with a 500 iteration window, and the last iteration of a single RWM inversion are shown in Fig. 6. Figure 7 shows the means, standard deviations, and differences with BedMachine of the three different ensembles.



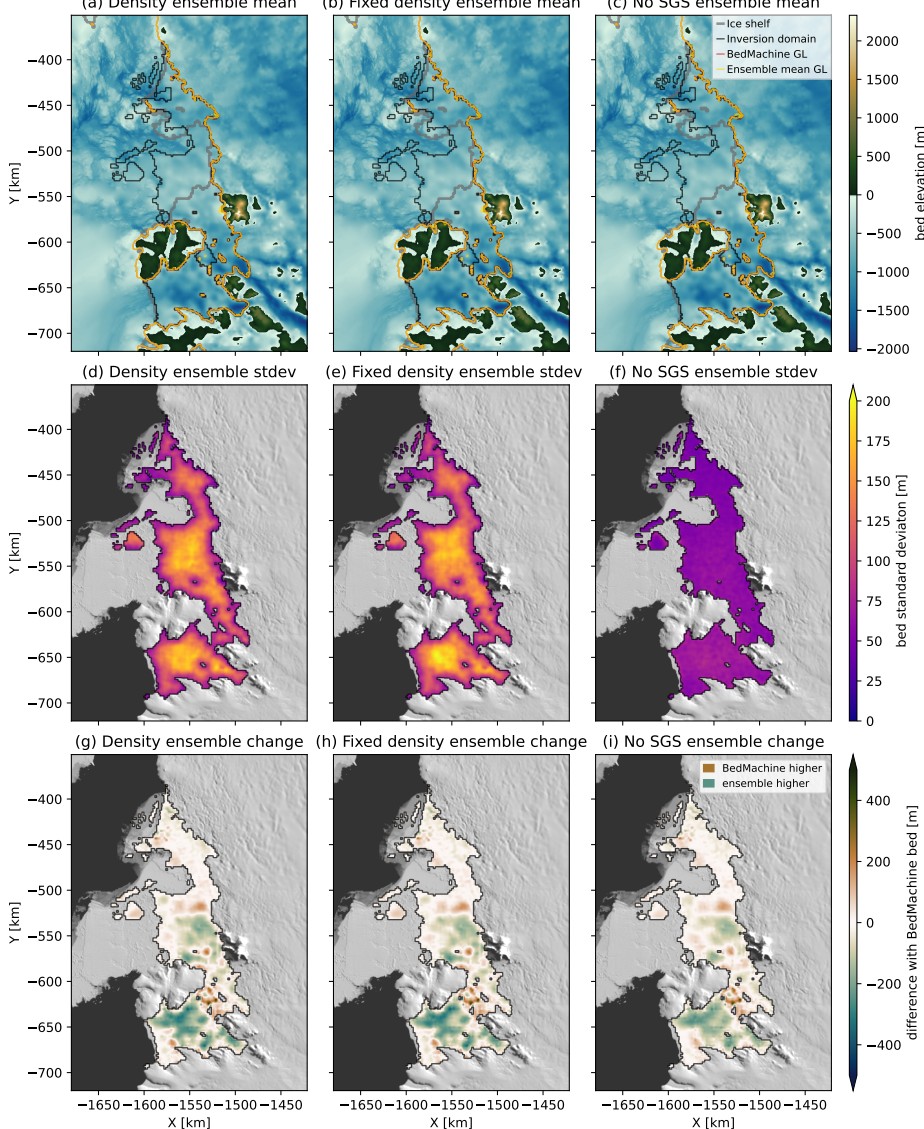

**Figure 7.** (a-c) Mean of bathymetry models in ensemble with variable background density, ensemble with fixed background density, and ensemble with deterministic interpolation. The BedMachine grounding line is shown in dark orange, and the ensemble mean grounding lines are shown in yellow. (d-f) Standard deviation of bathymetry models in same ensembles as row (a-c). (g-i) Difference with BedMachine v3 bed for same ensembles as rows above.

The first ensemble includes the SGS stochastic interpolation of the Bouguer disturbance and sampling background densities, the second ensemble includes SGS interpolation of the Bouguer anomaly but with constant background density of $2670 \, \mathrm{kg/m^3}$, and the final ensemble uses the deterministic ordinary kriging interpolation of the Bouguer disturbance. Each ensemble tested





produces the same general pattern in the mean bathymetry shown in Fig. 7a-c. The standard deviation of the ensembles are shown in Fig. 7d-f. The two ensembles including the SGS interpolation of the Bouguer disturbance (d and e in Fig. 7) have much higher uncertainty with values approaching zero near the boundary and reaching 200 m in the center of Dotson ice shelf, the deep cavity below Kohler Glacier, and the area offshore of the Crosson calving front. Conversely, the ensemble with the fixed Bouguer disturbance has a much lower standard deviation as seen in Fig. 7f. The similarity in the mean and standard deviation of the ensembles with SGS interpolation with and without sampling the background rock density shows that the rock density is not an important contributor to uncertainty but the SGS interpolation is. The differences between the ensemble means and BedMachine are shown in Fig. 7g-i. Large differences of over 200 m exist in the cavity below Kohler Glacier, the eastern edge of Bear Island, much of Dotson ice shelf, and the area between Crosson and Dotson ice shelves. In general, the ensemble means are shallower than BedMachine below most of Dotson ice shelf and Kohler Glacier, deeper than BedMachine in the area between Crosson and Dotson ice shelves, shallower near the terminus of Crosson ice shelf, and deeper than BedMachine near the Thwaites Glacier Tongue. The mean of the ensemble with no SGS interpolation is a slightly smoother version of the means of the ensembles with SGS interpolation. This is clear in the differences with BedMachine where the same general patterns exist but there are less pronounced differences. The final iteration from three random inversions from the first ensemble, representing different realizations of bathymetry, are shown in Fig. 8a-c. These individual inversion results have larger differences with BedMachine (Fig. 8d-f) than the ensemble means. Cross sections of the ensemble members and the ensemble mean from the ensemble without SGS are shown in Fig. 9. Cross sections from the ensemble with SGS are shown in Fig. 10.

## 5    Discussion

### 5.1    Implications of new ensembles

Gravity inversions provide a means to infer the bathymetry beneath ice shelves, which is critical for understanding how mCDW moves beneath ice shelves and drives melting. However, gravity inversions are fundamentally non-unique and rely on assumptions about the background density and the Bouguer disturbance, or the portion of the gravity disturbance which comes from crustal density variations and the thickness of the crust. Previously, the Bouguer disturbance has been interpolated using one-time, deterministic interpolations. Instead, our new approach uses SGS to sample the uncertainty of the Bouguer disturbance interpolation. Our annealed RWM approach is able to efficiently solve the inverse problem, enabling us to construct ensembles of bathymetry models. We produce three ensembles with 100 MCMC inversions each of the bathymetry beneath the Dotson, Crosson, and Thwaites Eastern Ice Shelf in the Amundsen Sea. In the ensembles including SGS interpolations, each inversion includes an independent realization of the interpolation of the Bouguer disturbance. Our MCMC approach to solving the inverse problem uses block updates such that each inversion samples the uncertainty of the gravity data and the nonuniqueness of the problem. The annealed block update schedule has a clear effect on the efficiency of the inversion (Fig. 6), with increases in acceptance rate after changing the parameters.




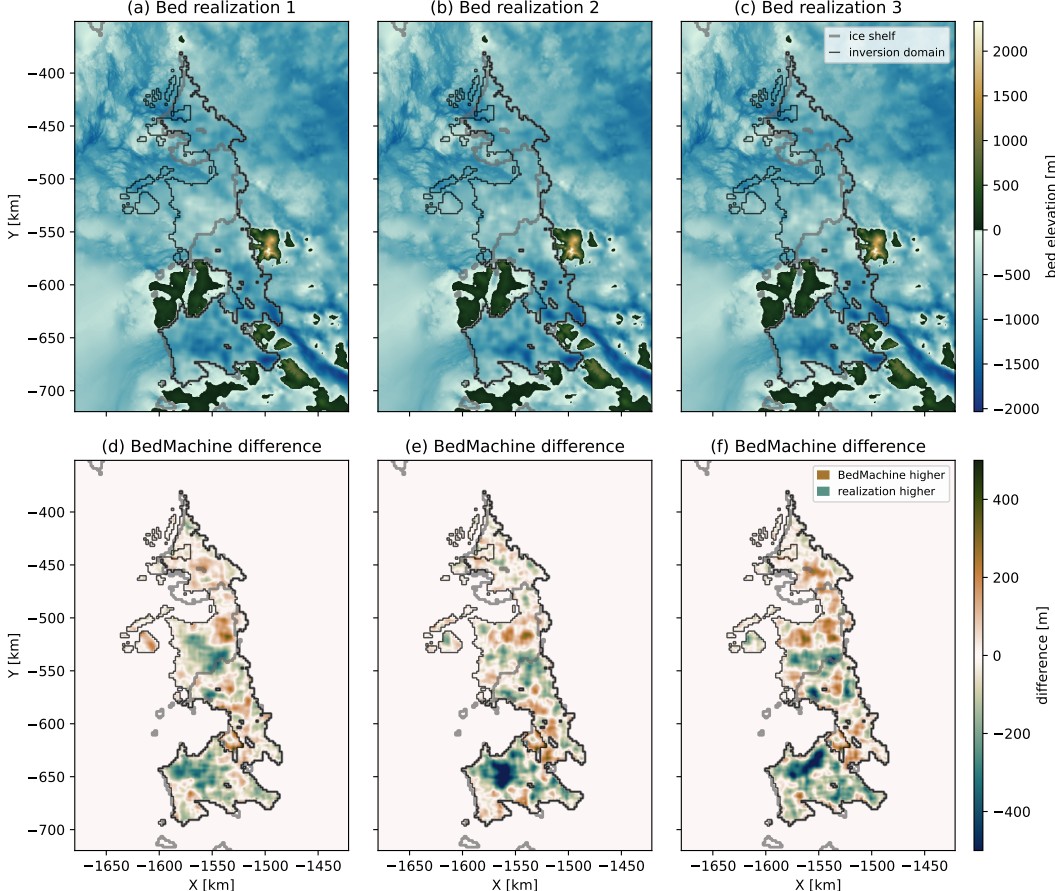

**Figure 8.** (a-c) Three different bathymetry models from the ensemble with SGS and variable background density. (d-f) Difference between BedMachine and the bed realization.

Through our set of three ensembles shown in Fig. 7, we can see that the Bouguer interpolation is the largest contributor to the uncertainty but the background density is not. This is shown by the fact that the ensemble with both Bouguer SGS interpolation and random background densities produces little difference with the ensemble with Bouguer SGS interpolation but a fixed background density of 2670 $\mathrm{kg/m^3}$. The final ensemble that uses a fixed density and a fixed mean Bouguer interpolation shows much smaller uncertainty, reflecting only the uncertainty due to crossover errors in the gravity data. The effect of the SGS interpolation of the Bouguer disturbance on the uncertainty is clearly visible in the difference between Fig. 9 and Fig. 10. These results show that uncertainty due to the background density may have been over-represented while uncertainty due to unknown geologic factors was unaccounted for. The previous inversions of Jordan et al. (2020b) and Tinto and Bell (2011) acknowledge that unknown geologic factors are difficult to account for. Our approach improves this issue by producing different bathymetry models that sample the uncertainty in the geology by stochastically interpolating the Bouguer disturbance.



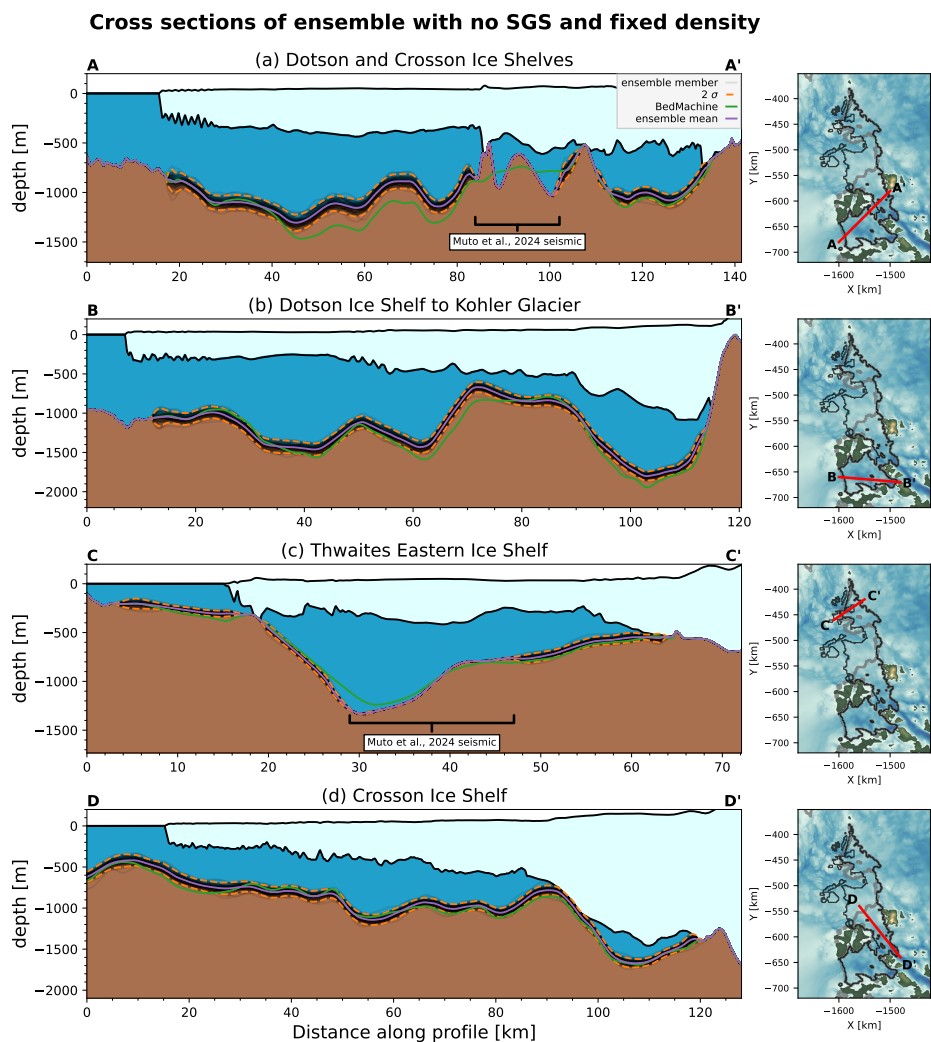

**Figure 9.** Cross sections from the ensemble with fixed mean Bouguer disturbance. The individual ensemble members are shown with transparent black, the ensemble mean with purple, the mean plus and minus two standard deviations with orange, and the BedMachine bed from Jordan et al. (2020b) in green. The plan view location of the the cross section is shown in the map to the right of each cross section. (a) Cross section of Dotson Ice Shelf to Crosson Ice Shelf. (b) Cross section of Dotson ice shelf including Kohler Glacier. (c) Cross section of Thwaites Eastern Ice Shelf (d) Cross section of Crosson Ice Shelf.

As such, we believe our approach represents the most robust inference and uncertainty quantification of the bathymetry beneath the Crosson, Dotson, and Thwaites Eastern ice shelves to date.

Our ensembles show consistent differences between the BedMachine bathymetry, which may have impacts on ocean circulation between the Crosson and Dotson ice shelves. Fig. 7g-i shows that each ensemble produces a mean bathymetry that
is generally deeper than BedMachine in the strait between the Crosson and Dotson ice shelves. This is because a portion




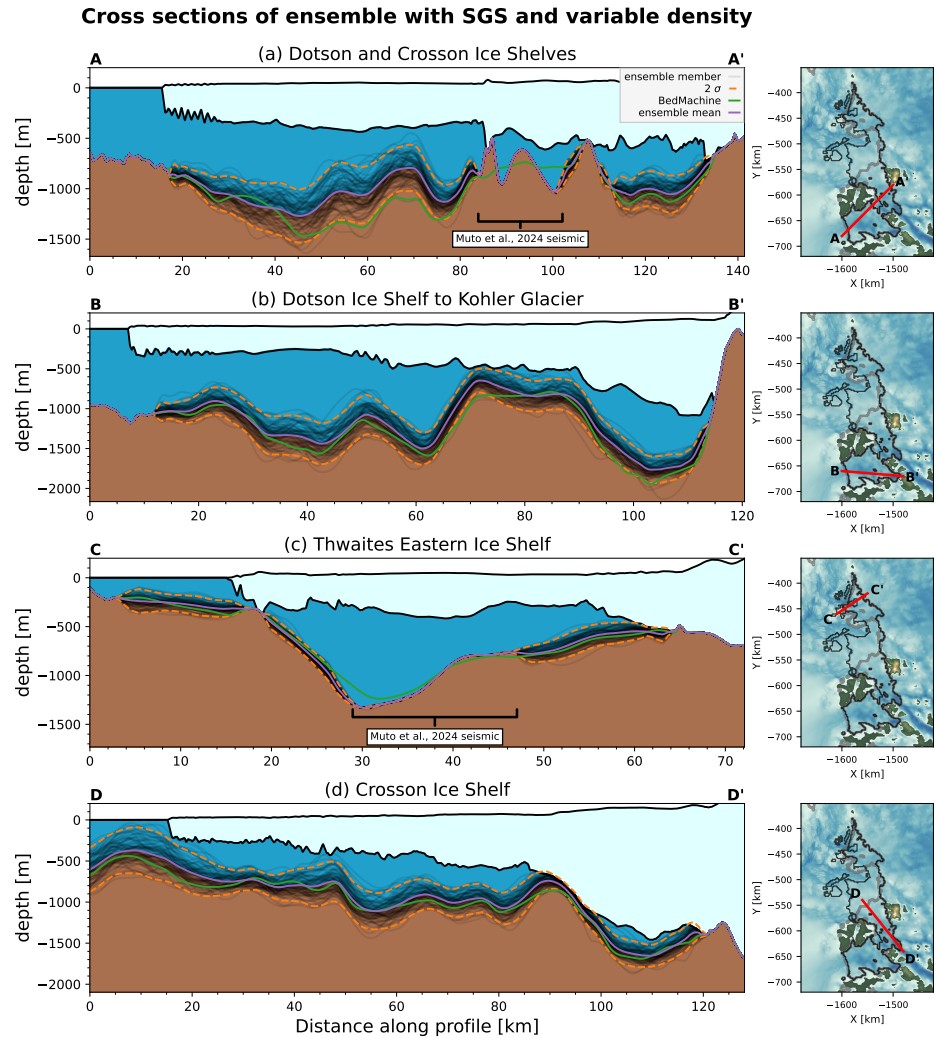

**Figure 10.** Cross sections from the ensemble with SGS interpolation of the Bouguer disturbance and variable background density. The individual ensemble members are shown with transparent black, the ensemble mean with purple, the mean plus and minus one standard deviation with red, the mean plus and minus two standard deviations with orange, and the BedMachine bed from Jordan et al. (2020b) in green. The plan view location of the the cross section is shown in the map to the right of each cross section. (a) Cross section of Dotson Ice Shelf to Crosson Ice Shelf. (b) Cross section of Dotson ice shelf including Kohler Glacier. (c) Cross section of Thwaites Eastern Ice Shelf (d) Cross section of Crosson Ice Shelf.

of the strait is constrained by seismic soundings from (Muto et al., 2024), which showed deeper seafloor than BedMachine with the exception of a notable rise in the center of the seismic sounding area. The deeper seafloor measurements from the seismic soundings then drives the gravity inversion to be similarly deeper than BedMachine around it due to the influence of the Bouguer interpolation. A deeper strait could mean that there is more transport of water between the eastern and western





portions of the Amundsen Sea than previously thought. The ability to move more water through this strait could indicate that more mCDW is delivered to the grounding line in this area than previously thought. Conversely, the mean of the ensembles are generally shallower than BedMachine below the Crosson and Dotson ice shelves, which could impact the amount of mCDW able to circulate in the sub-ice-shelf cavities. The means of the ensembles show little difference with BedMachine beneath Thwaites Eastern Ice Shelf. Individual models seen in Fig. 8 or Fig. 10 have much more variable water-column thickness than

the mean models, meaning the individual models could impede the movement of water more than the smoother representative models because of their more complex geometries. The variability seen in the SGS ensemble members is justified by the fact that the seismic constraints on Dotson Ice Shelf show large deviations from the BedMachine bathymetry (Fig. 10a). As such, our ensemble members capture the uncertainty in such bathymetric features below the ice shelves. It will be useful for ocean modelers to see to what extent the roughness of the sub-ice-shelf cavities influences circulation and how this differs from a

smoother model like the mean of an ensemble.

    Our ensembles of bathymetry models could be useful for exploring glaciation history. The variability of our models in the ensembles including SGS is such that they include bathymetric highs which could have previously pinned the ice shelves. These possible pinning points could help explain past advance and retreat behavior. Conversely, ice-sheet-model spin up could be used to narrow the ensemble of bathymetry model to only those that can reproduce present conditions. As such, the ensemble

of bathymetry models are useful for testing hypotheses about past ice-sheet configurations as well as further constraining the bathymetry.

    Our ensembles not only satisfy the goal of quantifying uncertainty across the inversion domain, but allow for easy propagation of the uncertainty in bathymetry to ocean and ice-sheet models. This could be accomplished by running ocean and ice-sheet models with different bathymetry realizations, thereby producing an ensemble of model outputs. In this way, the

uncertainty in the sub-ice-shelf melt and grounding-line retreat due to bathymetry can be better assessed. Given the computational cost of these downstream models, ensemble approaches to uncertainty quantification are not always feasible. However, recently both Rosier et al. (2023) and Burgard et al. (2023) demonstrated the use of deep learning to emulate an ocean model, thereby immensely reducing the computational cost of producing ice-shelf melt rates. Emulators like those could be used with our bathymetry ensembles to rapidly propagate the uncertainty in bathymetry to ice-shelf melt rates and ice loss projections.

As such, our stochastic approach to bathymetry inversion enables improved uncertainty quantification of bathymetry but also for future ice-sheet stability.

## 5.2   Limitations and future work

The nonuniqueness of the the gravity method is such that the sediment thickness cannot be estimated without sediment-thickness constraints from boreholes or seismic measurements. One advantage of our method is that the Bouguer disturbance

contains the gravity disturbance from density contrasts due to sedimentary rock, unconsolidated sediments, or dense igneous bodies. Therefore, despite not modeling sediment thickness directly, the SGS interpolation of the Bouguer disturbance can be interpreted as a stochastic interpolation of crustal density anomalies. However, some of the Bouguer disturbance in the conditioning area could arise from errors in the ice thickness from BedMachine as well as the gridding used to coarsen BedMachine





to a 2-km grid. In order to account for the uncertainty in the conditioning data, the bed data (e.g. radar and sonar data) could

be sampled from a distribution at each iteration in the chain instead of being treated as a deterministic value. Our inversion approach could also be improved by refining the grid near the grounding lines and pinning points, both of which exhibit complex geometries (Rignot et al., 2014). While the gravity inversion could not necessarily resolve the bathymetry at these finer resolutions, the complex structures that are important to the ice-shelf dynamics could be better resolved. Furthermore, the annealing schedule used for the block updates was chosen arbitrarily through testing, and more systematic approaches to choosing the

block update parameters could be used. Alternatively, the block size, correlation distance, and amplitude could all be treated as random variables that are sampled from each iteration, giving the benefits of larger and smaller updates throughout the whole inversion. Many such improvements should be considered in the future but ultimately, the combination of conditional simulation and stochastic optimization to interpolate the Bouguer disturbance and produce ensembles of bathymetry models is a step forward in incorporating the uncertainty due to unknown geologic factors in the inversion.

Our inversion uses only the vertical component of the acceleration due to gravity, but would greatly benefit from having direct measurements of the gradient of the gravity field. A test study from Yang et al. (2020) demonstrated that the inversion of bathymetry benefits from the inclusion of gravity-gradient data. Additionally, gravity-gradient data could improve the inversion of subglacial geology because of increased sensitivity to lateral changes in the gravity field. For the same reasons, the petroleum industry has routinely used gravity-gradient data to improve the delineation of subsurface oil and gas reservoirs

(e.g., Oliveira and Barbosa, 2013; Paoletti et al., 2016). Because of this, it would be beneficial for the community to explore future opportunities to acquire full-tensor gravity-gradient data.

We chose to study the Dotson, Crosson, and Thwaites Eastern Ice shelves because of their sensitivity to mCWD moving up the continental shelf and the important role they play in buttressing the fast-flowing glaciers terminating in the Amundsen Sea (e.g., Scambos et al., 2017; Jenkins et al., 2016). However, our method is readily transferable to other ice shelves in

Antarctica and Greenland. Our approach requires airborne gravity data to be present and benefits from dense RES data on grounded ice and MBES or seismic bathymetry measurements offshore. The Thwaites region has dense airborne gravity data relative to other ice shelves. Where airborne gravity data is sparse, satellite gravity data can be used to fill in gaps. The ANTGG2021 continent-wide gravity product uses a combination of airborne and satellite gravity data, and can be used at ice shelves where airborne data is sparse (Scheinert et al., 2016; Charrassin et al., 2025). At other ice shelves, it may be necessary

to detrend the Bouguer disturbance before odin SGS interpolations. In addition, directional variograms and anisotropic SGS interpolations could be used in cases where there are elongated features in the Bouguer disturbance conditioning data. The fewer bed measurements there are, the less conditioning data are available to calculate the variogram of the Bouguer disturbance and the greater the uncertainty will be in SGS interpolations. Our approach motivates the continued acquisition of airborne and seaborne geophysical measurements along the Antarctic coast to improve bathymetric estimates and uncertainty quantification.

The uncertainty derived from our bathymetry ensembles could be instructive in planning future underwater autonomous and remotely operated vehicle missions, seismic soundings, or further airborne geophysics to areas of high uncertainty (see e.g., Schmidt et al., 2023).





## 5.3 Application of workflow to other problems

Our stochastic approach to solving the gravity inverse problem could be utilized to improve the inference and uncertainty
quantification of other subglacial conditions like subglacial topography and geology. The benefit of this approach is that it
is flexible and can accommodate any data or physical relationship that can be formulated into an optimization problem. For
instance, our approach could be modified to simulate realistically rough topography that conserves ice mass in ice-sheet models.
Besides physics constraints, various hard constraints, such as new grounding lines or better constrained pinning points can
easily be added in our approach. While the optimization framework is easily transferable, the method relies on a fast forward
model to be effective, and the block update schedule must be tested to ensure efficient convergence and to prevent overfitting.
The application of our approach to other subglacial problems would further enable improved uncertainty quantification of those
subglacial conditions and the ice-sheet models that depend on them.

Our approach can also be adapted to other inversion problems in the geosciences. Many problems are nonlinear and in-
clude interpolations. Our workflow enables uncertainty quantification of parameter fields which have impacts on processes or
observations, such as hydraulic conductivity, density, and magnetic susceptibility. First, SGS simulation of spatial fields can
generate stochastic realizations of the spatial field while preserving the spatial statistics of the conditioning data. Next, our
MCMC approach can be paired with a forward model to efficiently perturb a surface or multiple surfaces in an earth model to
fit the forward modeled response to the observed data. Our approach could also be extended from perturbing a surface with 2D
blocks to perturbing a volume using 3D random fields. This could be used to invert for a 3D density distribution using gravity,
for example. The combination of the SGS simulation and MCMC stochastic optimizations means that the geostatistical prior
uncertainty can be transformed into a posterior uncertainty of the response variable. Thus, our approach provides a simple a
flexible uncertainty quantification framework that may be of interest in a variety of use cases in the geosciences.

## 6 Conclusions

We have presented three new ensembles of bathymetry models for the Crosson, Dotson, and Thwaites Eastern ice shelves
which allowed us to quantify the uncertainty in the bathymetry due to the interpolation of the Bouguer gravity disturbance,
the uncertainty in the gravity data, and the uncertainty in the background density. The three different ensembles show that the
background density is not an important contributor to uncertainty in the bathymetry while the SGS interpolation is the largest
source of uncertainty. Our different bathymetry models differ greatly from the widely used BedMachine v3 bathymetry and
the mean models of each ensemble show consistent differences with BedMachine. Our ensembles can be used to propagate the
uncertainty in bathymetry to ocean and ice-sheet models to quantify uncertainty in ice-shelf melt and future stability.

We developed our ensembles by solving the gravity inverse problem using a Random Walk Metropolis MCMC approach
which uses successively refined block updates to alleviate the difficulties imposed by the high dimensionality of the problem.
Our approach only computes the gravity for the prisms which were perturbed, speeding up the forward gravity calculation
such that each inversion in the ensemble is completed in about 5 minutes. Our approach highlights the utility of MCMC in



estimating subglacial geometry and subglacial conditions constrained by geophysical data, and could be adapted to explore other subglacial conditions.

*Code and data availability.* The code and bathymetry ensembles are archived on Zenodo (https://doi.org/10.5281/zenodo.15258019; Field, 2025). The code is available on GitHub under MIT license (https://github.com/mjfield2/stochastic_bathymetry). OIB gravity data is can be explored on https://nsidc.org/icebridge/portal/map (Tinto et al., 2010). ITGC gravity data can be downloaded from the British Antarctic
Survey (Jordan et al., 2020a). BedMachine v3 is available from the National Snow and Ice Data Center (NSIDC) (Morlighem, 2022). MODIS Mosaic of Antarctica 2014 is available from the NSIDC (Haran et al., 2017). Phase-based ice velocity is available from the NSIDC (Mouginot et al., 2019b). Grounding zone shape files are available from the NSIDC (Rignot et al., 2023). Thwaites and Dotson seismic sounding data are available from the U.S. Antarctic Program Data Center (Muto et al., 2024). This work used the publicly available Python packages GStatSim (MacKie et al., 2023), Harmonica (Fatiando A Terra Project et al., 2023), Verde (Uieda, 2018), SciKit-GStat (Mälicke et al., 2022), and
GSTools (Müller and Schüler, 2023). All figures were made with Matplotlib version 3.10.0 (Team, 2024; Hunter, 2007).

*Author contributions.* M. J. F. contributed to project design, software development, and writing. E. J. M. contributed to project conceptualization, project design, and writing. L. W., A. M., and N. S. contributed to writing and project design.

*Competing interests.* The authors declare no competing interests.

*Acknowledgements.* We thank Mareen Lösing and Kirsty Tinto for helpful comments. We thank the Fatiando a Terra team for creating the
open source software packages Harmonica and Verde, which played central roles in this work. M. J. F. and E. J. M. were supported by NSF award 2324092. A.M. was supported by NSF award 1929991.





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
