# Peer review of "Improved Bathymetry Estimates Beneath Amundsen Sea Ice Shelves using a Markov Chain Monte Carlo Gravity Inversion (GravMCMC, version 1)"

_EGUsphere, 2025_

## Author Comment (AC1)

RC1:

Thank you for the thoughtful review. We agree that including gravity equations and a visual will help the reader understand the different components of the gravity disturbance. The reviewer is correct in understanding that the MCMC is not used in a traditional way, where thousands or millions of samples from an MCMC chain are used to quantify uncertainty, and the uncertainty is driven by the SGS interpolation of the Bouguer disturbance. Indeed, the Random Walk Metropolis Hastings approach with block updates is used more as a stochastic optimization approach and a single sample from within the posterior distribution is taken from each inversion for the ensemble. It is true that the results would be very similar if a different optimization approach, such as simulated annealing, were used. However, we do see advantages in using the approach that we did. The primary advantage is that the spread of the likelihood (s in equation 6) can be interpreted as measurement uncertainty, thus we are able to sample that uncertainty using this approach. However if we used simulated annealing, this parameter would be the temperature, and would be a tunable parameter that decreases such that the acceptable moves hone in closer and closer to a fit with the observed data. Regardless, we will prioritize making these points clear and de-emphasizing the term MCMC so as not to mislead the reader.

Specific comments

L10 'inversion is efficiently solved' -> 'inversion is efficiently computed?' On the grounds that 'inversion' is (slang for) a solution to an optimization problem, rather than the problem

     changed

L17 'Introduction and Background' -> delete

     changed

L19 'amount of melt' -> melt rate

     changed

L25 'grounding line retreat' -> ice dynamics? GL retreat is just one aspect of the ice flow response to ice shelf ablation.

     Changed to "grounding line retreat and ice shelf thinning" to not single out GL retreat as the only response to ice shelf ablation.

Discussion of the Bouguer disturbance (l45 on and L66 on) – see general comments, but also, the difference between anomaly/disturbance etc. Either 'disturbance' is an uncontroversial term – then use it without commentary – or not – in which case use the usual terms.

     Changed to only using disturbance terminology and removed the paragraph. Defined gravity disturbance in introduction.

Literature review starting L63, and running to about L130. I found some parts difficult to follow, with very long paragraphs, especially the first. The last two paragraphs were easier for me – each one makes a single point and backs it up with brief, reasonably detailed examples.

We have reworked this portion of the literature review, eliminating the long and overly-detailed paragraph in favor of focusing on the core gaps that we aim to address. These gaps are the lack of multiple realizations/solutions, the use of a deterministic interpolation of the Bouguer disturbance, and the lack of spatial uncertainty quantification in some studies.

L130 'One approach to generating multiple solutions is MCMC.' Yes, but that is not the source of multiples here (as it is in other places and notably Mosegaard and Tarantola). Correct? I don't think much change is needed here, just a different opening (MCMC methods have long been used in gravity inversion….)

Changed opening of paragraph

L185 – discussion of upward continuation – introduced mid-paragraph and not defined.

Improved explanation of what upward continuation is.

L208 – 100 inversions run in parallel on (a machine that has < 20 cores). State the wall time for a single MCMC

Clarified that each SGS and MCMC inversion has a combined wall time of about 5 minutes

Table 1 could be expanded to summarize the ensembles more completely, e.g ensemble 2 comprises 100 realizations, with realization n generated by selecting one item from an MCMC chain subject to the nth SGS realization etc.

Changed to have a more detailed description.

240 (and on). The choice to use MCMC as optimization procedure, rather than for sampling a distribution is unusual (not least because the rate of convergence is low). Some comment on reasoning would help. I imagine that the degree od autocorrelation in each chain is high, especially given the block updates, so that long chains would be needed. Ruggeri et al *Geophys. J. Int.* (2015) **202,** 961–975 have chains with ~10^6 iterations

Our approach is more similar to simulated annealing, which in its basic form is very similar to the Metropolis Hastings random walk. The primary differences between our approach and simulated annealing are that we maintain the *s* parameter as our measurement uncertainty, whereas in simulated annealing this would be the temperature that is gradually reduced. In simulated annealing the effect of reducing *s* is that the proposed jumps which do not improve the misfit become increasingly penalized until only proposals that improve the misfit are accepted. We maintained the *s* parameter as the measurement uncertainty so that we are pushing the model towards samples of the bathymetry that satisfy the gravity measurements

within the measurement uncertainty. Thus, our approach is not too different from a commonly used stochastic optimization procedure. We would argue that the rate of convergence is not low, owing to the efficiency by which our approach can reduce the loss to within the prescribed stopping criterion (about 3-5 minutes). The block updates are designed to maintain a high degree of efficiency, with large blocks with long correlation lengths lowering the loss initially, followed by smaller blocks with shorter correlation lengths refining the details of the bathymetry.

Initially in this project we used Gaussian random fields over the whole domain to perturb all the model parameters at once. This resulted in very long autocorrelation lengths and the acceptance rate gradually going to 0. This motivated us to move to the block update approach which can maintain high acceptance rates and rapidly achieve a bathymetry model that is within distribution. With our current approach we are not using the chain in the traditional MCMC approach where we would generate a posterior from a chain. Thus, we are not concerned with the autocorrelation of the chains. Shown below is a trace plot for 3 locations within the inversion domain. Indeed it can be seen that there are autocorrelation lengths many times longer than the chain but the grid cells have generally moved from the initial condition to within distribution.

[Figure]

Eqn 1. The notation \sigma(m) = \kappa \rho(m) L(m) is from Mosegaard and Tarantola, and I see others use it too. But more often Bayes rule is stated with more explicit reference to conditional probabilities, e.g p(m|d) = p(d|m) p(m) / p(d). Where p(d|m) = L(m), but now makes clear that the likelihood is the probability of the observed data given the model, and so on. I would also say that the prior cannot be ignored – rather you have chosen a uniform distribution p(m) between some bounds, which means that p(m | d) i= \kappa p(d|m) within said bounds.

Changed to more accurately show the conditional probabilities of Bayes' Theorem. Added sentence about the prior distributions being uniform and how this leads to their cancellation.

L275 – seems like a strange way of saying 'we have a formula for sigma(m) in terms of L(m), derived just a few lines back'

Removed sentence

Table 2: define width and range.

Defined in table heading.

Figure 7 / results in general. How do you know that 100 ensemble members is sufficient. You could plot some summary quantities F(n). e.g. the sum-of-squares over the domain of the standard deviation where F(n) means the F computed from members 1 to n

It is our hope that our code can be viewed as a data generation process that can create a new bathymetry model in 5 minutes or less. Thus, if 100 realizations is insufficient for a particular use case more models can be realized relatively easily. Additionally, we believe that the main contribution of this work is in the ensemble members which can be used in ocean models and not necessarily in the standard deviation maps which are more so to show the general magnitude. The ocean models are ice sheet models that these bathymetry models may be used in are computationally expensive, so it is unlikely that more than 100 will be used for such analyses. However if more are needed, more can be generated. We produced the suggested figure and while there is some variation in the sum of squares for larger numbers of models, we believe the flatness of the curve above about 60 models shows that the standard deviation is stable.

[Figure]

Discussion section: is quite long – section 5.3 seems to not propose anything concrete. I do agree that the general framework could be applied elsewhere, but the key innovations e.g the SGS of Bouguer correction are domain specific.

Eliminated "applications to other work flows" section and replaced with a paragraph focused on how our approach could be adapted to magnetics data and density inversions.